# CleanPage: Fast and Clean Document and Whiteboard Capture

**DOI:** 10.3390/jimaging6100102

**Published:** 2020-10-01

**Authors:** Jane Courtney

**Affiliations:** School of Electrical & Electronic Engineering, Technological University Dublin, City Campus, Dublin, Ireland; jane.courtney@tudublin.ie

**Keywords:** document scanning, whiteboard capture, image enhancement, image alignment, image registration, image quality assessment, NR-IQAs

## Abstract

The move from paper to online is not only necessary for remote working, it is also significantly more sustainable. This trend has seen a rising need for the high-quality digitization of content from pages and whiteboards to sharable online material. However, capturing this information is not always easy nor are the results always satisfactory. Available scanning apps vary in their usability and do not always produce clean results, retaining surface imperfections from the page or whiteboard in their output images. CleanPage, a novel smartphone-based document and whiteboard scanning system, is presented. CleanPage requires one button-tap to capture, identify, crop, and clean an image of a page or whiteboard. Unlike equivalent systems, no user intervention is required during processing, and the result is a high-contrast, low-noise image with a clean homogenous background. Results are presented for a selection of scenarios showing the versatility of the design. CleanPage is compared with two market leader scanning apps using two testing approaches: real paper scans and ground-truth comparisons. These comparisons are achieved by a new testing methodology that allows scans to be compared to unscanned counterparts by using synthesized images. Real paper scans are tested using image quality measures. An evaluation of standard image quality assessments is included in this work, and a novel quality measure for scanned images is proposed and validated. The user experience for each scanning app is assessed, showing CleanPage to be fast and easier to use.

## 1. Introduction

In the worldwide COVID-19 pandemic 2020, many faced a sudden thrust into an online-only working environment, making paper document sharing and physical whiteboard usage impossible. In addition to this move to an online-only environment, many documents still exist only in paper form and must be scanned to digitize them. Many users also enjoy the “chalk-and-talk” appeal of whiteboards but are left with no record of their work on erasure. With the ubiquity of smartphones and a variety of scanning apps available, digitization is becoming a simpler task.

However, in most scanning apps, the output quality of the scans is highly dependent on the quality of the one image captured and on the performance of the user in the capture process. As the camera is handheld, issues such as motion, focus, distance, lighting variations and perspective distortion are to be expected and have significant impacts on the success of the scan. To overcome these, most scanning apps offer additional manual intervention steps, such as corner adjustment or lighting correction. This has an impact on the user experience and slows scanning time considerably. The results are also affected by the quality of the paper or whiteboard surface and by clutter in the background of the image. In all the scanning apps tested, the background surface was not corrected, leading to paper or whiteboard imperfections appearing in the final output.

Document scanning now focuses on the detection of documents in images [1,2,3] and content recognition [4,5]—typically Optical Character Recognition (OCR) for text—but the problem of high-quality capture has not been fully solved. Available apps such as Microsoft Office Lens, Google PhotoScan, Cam Scanner, Adobe Scan, Scanbot and others produce high-quality results but are sensitive to capture conditions, retain surface imperfections, require manual intervention, and often have slow or difficult to use user interfaces.

For input image capture, many scanning approaches use image stitching or mosaicking [6,7], which requires moving the camera around the document or whiteboard in a controlled way [8]. Others use deep learning techniques to find the object in the image in order to limit the capture area to content [9,10]. Improvements on these methods often focus on estimating text direction [11], limiting their usability to text-only images.

In the area of whiteboard capture, surprisingly little work has been done on whiteboard image enhancement, despite poor performance of existing methods on dirty whiteboards [12,13]. Though early work in the area has developed into high-quality apps, e.g., Zhang and He’s work [14], which ultimately contributed to the development of Microsoft Office Lens, these solutions still suffer from the same dependency on high-quality input images and therefore on user performance [15]. A review of existing whiteboard enhancement methods can be found in Assefa et al. [16].

When it comes to document scanning, document image enhancement is still an open field of research, with most methods focusing on document binarization [17,18,19,20]. The problem of document binarization, i.e., moving toward a binary or black-and-white-only image, is still unsolved, and a comprehensive review of the challenges involved can be found in Sulaiman et al. [21]. By 2015, the document scanning problem was still not fully solved, and the SmartDoc competition [22] was established to encourage development in the area and to evaluate the performance of document imaging methods. In the 2017 version [23], it was recognized that the solution may lie in video-based scanning. However, it was assumed that any solution would use stitching or mosaicking, and so the videos in the provided dataset were captured using the usual input method, i.e., by moving the camera around the document in a controlled way.

Some multi-image or video-based approaches to scanning have been offered for the problem. Zhukovsky et al. [24] use a segment graph-based approach to find the document boundaries, while Chen et al. [25] use video as a guide to the user before static image capture. Jiang et al. [26] use key image extraction and image stitching for capture and Retinex processing [27] for enhancement. Luqman et al. [28] combine mosaicking with data from the phone’s own accelerometer to improve results from the usual moving-camera input method.

To date, no solution has been found that retains a fast and simple user interface while addressing the limitations of existing scanning apps, namely: surface imperfections, background clutter, and sensitivity to both capture conditions and user performance.

## 2. Materials and Methods

In CleanPage, the user interface (UI) is simplified to a single button. Instead of capturing a single image, it records a short sequence of images, similar to the operation of High Dynamic Range (HDR) photography and super-resolution techniques [29]. However, unlike HDR and super-resolution, this system includes an alignment phase to overcome shortcomings in user capture and an enhancement phase to remove background surface imperfections. The system works by calculating inter-image motion vectors and homography to align and stack the images before merging these to a single output. Then, this is processed to enhance contrast and remove any background clutter by cropping to content. The output is a high-contrast, low-noise image with an ideal homogenous background surface. Since the input is a set of images instead of a single image, the system is less reliant on user performance, and high-quality results are consistent across a variety of scan types. CleanPage’s design results in improved background homogeneity, improved content retention, and a simple and fast UI.

### 2.1. CleanPage Design

This system was designed in Python 3.7 with OpenCV 4.3 and ported to an Android app developed in Kivy using Buildozer. Testing was done on a PC using MATLAB 2020b for image quality tests.

#### 2.1.1. Image Capture

In CleanPage, images are acquired as a short sequence with one button-tap by the user. The raw RGB data is split into its three color channels, and each is handled separately during the merging phase. The original image is also converted to a grayscale image, *I* (the intensity or average of the three channels), for use in the alignment phase.

#### 2.1.2. Alignment

Once the images are acquired, these are aligned before merging. Alignment is achieved by first extracting distinct features from each image. Then, these features are tracked to the next image by determining the direction and magnitude of motion from image to image. The motion vectors extracted allow the homography between images to be computed. This is used to warp the images to an aligned position.

To find distinct features in an image, it is important that the method is scale, rotation, and illumination invariant, as hand motion is expected through the image sequence. There are many feature descriptors available, ranging from SIFT-based keypoints (SIFT, SURF, KAZE, ORB, etc. [30]) to simple corner detectors [31]. Here, Shi-Tomasi features are used as they are quick to compute and suitable for tracking, while remaining robust to scale, rotation, and illumination changes [32].

Shi-Tomasi features are determined from the autocorrelation matrix:(1)$$A=\underset{i}{\sum }\underset{j}{\sum }g\left(i,j\right)\left(\begin{array}{cc} I_{X}^{2} & I_{X}I_{Y}\\ I_{X}I_{Y} & I_{Y}^{2} \end{array}\right)$$
g(i, j) is a set of Gaussian weights defined over a neighborhood region and $I_{X}$ and $I_{Y}$ are the horizontal and vertical gradients respectively of the intensity image, *I*.

Then, the feature quality measure is the smallest eigenvalue of *A*:(2)$$F=\min\left(\lambda_{1},\;\lambda_{2}\right)$$
*F* is the feature measure, and $\lambda_{1}$ and $\lambda_{2}$ are the eigenvalues of *A*. The Shi-Tomasi algorithm also includes non-maximal suppression to avoid the “clumping” of features and allows other parameters to be set, such as the minimum distance between features and the maximum number of features.

As the motion between images is small, simple optical flow can be used to determine the motion vectors. Here, Lucas-Kanade optical flow is used [33]. This is similar to the feature extraction method outlined above, but uses the brightness constancy assumption to incorporate the time dimension. The brightness constancy assumption states that the appearance of a feature does not change significantly from image to image when the motion is small. This means that the motion can be determined by a pair of horizontal and vertical vectors, (u,v), such that:(3)I(x,y,t)≈I(x+u,y+v,t+1)

Then, these motion vectors can be extracted using the autocorrelation matrix from before:(4)$$\left(\begin{array}{c} u\\ v \end{array}\right)=A^{-1}\left(\begin{array}{c} \underset{i}{\sum }\underset{j}{\sum }g\left(i,j\right)I_{X}I_{T}\\ \underset{i}{\sum }\underset{j}{\sum }g\left(i,j\right)I_{Y}I_{T} \end{array}\right)$$
$I_{T}$ is the intensity gradient in the time direction (from image to image), while g(i,j) represents the parameters of a 2D window of Gaussian weights, and $I_{X}$ and $I_{Y}$ are the horizontal and vertical gradients respectively of the intensity image, *I*, as before.

Using the calculated motion vectors, the location of corresponding features from each image can be found in the next image. This creates a set of $\left(x_{t},y_{t}\right)$ locations of the features in each image, where t denotes the image number.

As the camera is handheld, it moves slightly during capture. This creates a different image plane for each image in the sequence. In order to re-align the images, the set of corresponding feature locations are used to compute the 2D homography between the images:(5)$$\left(\begin{array}{c} x_{t}\\ y_{t}\\ 1 \end{array}\right)=H\left(\begin{array}{c} x_{t+1}\\ y_{t+1}\\ 1 \end{array}\right)$$

The transformation between images as a result of camera movement is not generally considered a homography. However, in this case, the feature points correspond to the same planar surface (the page or whiteboard), so the relationship holds true. This homography is calculated using the robust estimation method, RANSAC (RANdom SAmple Consensus) [34]. Once the homography, H, has been estimated, it can be used to affine warp the entire image to align with a reference image, so that all the pixels align with their corresponding pixel in each image. This homography-based method could also be used to correct perspective distortion, as the images can be aligned to a fixed, pre-defined orientation.

#### 2.1.3. Merging

After the images have been aligned, they are merged into a single image, *M*. This is done on each color channel separately by calculating the median of each pixel’s color value across the entire image sequence:(6)$$M\left(i,j\right)=\left(\tilde{R}\left(i,j\right),\;\tilde{G}\left(i,j\right),\tilde{B}\left(i,j\right)\right)$$
where $\tilde{R},\;\;\tilde{G}\;\mathrm{and}\;\tilde{B}$ are the medians of the red, green, and blue color channels of pixel (i,j) respectively across the image sequence. This method has the added advantage of minimizing any noise in the images, as each pixel represents the median value of that pixel over a set of images of the same content.

#### 2.1.4. Border Removal

As a result of the affine warping, some empty space is introduced near the borders of the image where no corresponding pixels exist. This is to be expected and appears in the aligned images as a black border. This removes some of the background clutter and provides a starting point for finding the background surface (paper or whiteboard). 

In order to identify regions of background surface in the image, the intensity is first scaled to enhance the background surface color. To do this, the Yen threshold [35], y, is calculated, and the image intensity is scaled from range [0, y] to the full range [0,255]. Then, this is thresholded to extract regions of background surface. The resultant image is a binary mask, B, with the surface represented by a zero value. 

To remove elements of background clutter introduced by the user’s inaccurate capture, a search box is initiated at the edge of the black border produced by the alignment phase. Each side of this search box is contracted, and the sum of its pixels from the binary surface mask, B, is calculated at each contraction. When this sum reaches a minimum value, it is determined that the location is entirely on the background surface, and then the image is cropped to this point. The locations of each side are found by calculating the sum of binary pixels at each row and each column for every nth contraction: (7)$$\begin{array}{c} R\left(n\right)=\overset{j=r}{\underset{j=l}{\sum }}B\left(n,j\right)\;n\;\in \left\{0\ldots h\right\}\\ C\left(n\right)=\overset{i=b}{\underset{i=t}{\sum }}B\left(i,n\right)\;n\;\in \left\{0\ldots w\right\} \end{array}$$
where R(n) is the pixel sum at the nth row and C(n) is the pixel sum at the nth column from the edge, t,b,l and r are the top, bottom, left, and right locations, updated recursively, and w and h are the width and height of the image, respectively. The minimum value of *n* that gives a minimum value for R is found to be the topmost surface row, t, while the maximum value is the bottommost surface row, b. Similarly, the leftmost and rightmost locations, l and r, are found from the column sums.
(8)$$\begin{array}{c} t=\min(\underset{n}{\mathrm{argmin}}\left(R\right))\;\\ b=\max(\underset{n}{\mathrm{argmin}}\left(R\right))\;\\ l=\min(\underset{n}{\mathrm{argmin}}\left(C\right))\;\\ r=\max(\underset{n}{\mathrm{argmin}}\left(C\right))\; \end{array}$$

In other border removal designs, contours are generally used to find the edges of the surface. However, this requires the entire surface to be visible in the image, and it must have a strong contrast with the background. By using the method outlined here, the system is not limited by these constraints.

#### 2.1.5. Image Enhancement

In the final stage of the design, the image is enhanced using two improvements: a sharpening kernel to remove blurring in the content area and intensity scaling to improve contrast in the background surface.

The merging phase causes significant reduction in noise, but this can also cause some blurring, and so sharpening is applied after merging to correct this. To avoid the reintroduction of noise, the sharpening is applied only in the region of content by using a dilated version of the binary mask, *B*, which was calculated previously in the border removal phase. The following sharpening kernel is used:(9)$$k=\left(\begin{array}{ccc} -1 & -1 & -1\\ -1 & \;9 & -1\\ -1 & -1 & -1 \end{array}\right)$$

To enhance the color of the background surface, the Yen threshold [35], y, is recalculated and the image intensity is rescaled in this region. This recalculation is done to improve the performance of the enhancement phase, since the absence of background clutter will improve the yen threshold calculation.

The outputs of each stage of CleanPage’s design can be seen in Figure 1, showing the process from the original images to the final cropped and enhanced image scan.

### 2.2. Data Inputs

To test the versatility and quality of the output, four scan styles were used: synthesized color test images, synthesized text test images, text on paper pages, and content on crumpled paper. Datasets of approximately fifty of each image type were produced. Samples of these can be seen in Figure 2. The synthesized test images (color and text) are high resolution, uncompressed, simple images, and they were displayed on a tablet and scanned with a smartphone. This novel testing method allows the original images to be used as ground truths. The real paper images were scanned in the normal way, using a smartphone. As these are on real paper, no reference will be available for comparison, so image quality measures were investigated for testing performance.

In all cases, images were produced by scanning with CleanPage as well as two of the leading scanning apps available: Microsoft Office Lens and Google PhotoScan. Other scanning apps were trialed, but it was found that they fall into the two categories of design represented by these two apps: single shot designs (Microsoft Office Lens) and stitching approaches (Google PhotoScan). All images were scanned in the same lighting conditions and background environment, and all were captured at the smartphone camera’s maximum resolution (12 megapixels).

### 2.3. Image Quality Measures

For the synthesized test image datasets, the originals are used as ground truths and a Structural Similarity score (SSIM) is calculated for each of the scanning methods compared with these images.

For the paper images, there is no ground truth for comparison, so a No-Reference Image Quality Assessment (NR-IQA) is needed. The most commonly used NR-IQAs were investigated: BRISQUE (Blind Referenceless Image Spatial Quality Evaluator [36]), NIQE (Natural Image Quality Evaluator [37]) and PIQE (psychovisually-based image quality evaluator [38]). However, in testing, it was found that these measures are not suitable for scanned images. Since they are designed to work with natural images, most NR-IQAs are in fact likely to favor the more natural, but less clean and accurate, photographic-style images over the ideal, homogenous background, which is the goal of a high-contrast scan.

Instead, for assessing the success of the scan, a measure of homogeneity is needed to test the consistency of the surface background. For this, the discrete entropy (DE) is used:(10)$$DE=\sum^{N}p_{I}\log_{2}p_{I}.$$

In a natural image, DE is seen as a measure of quality [39], as it represents the heterogeneity of the image. However, in a document or whiteboard image, it is expected that the background surface should be a large homogenous area (paper/whiteboard), so it is expected that in a high-quality image of this type, DE should be low. 

This has been shown previously [40] where natural images and text images were compared under increasing distortion levels. The natural image DE decreases with distortion, while the text image DE increases significantly. Entropy has also been used previously to improve binarization methods for text recognition for this reason [41].

## 3. Results

### 3.1. Assessment of Image Quality Measures

To test potential NR-IQAs, two datasets of approximately fifty synthesized reference images were produced and scanned, and the original unscanned versions were included in the assessment. As these originals are not scanned, they should represent higher quality and should give better scores in any image quality measure than scanned versions, regardless of the scanning app used. Standard NR-IQAs (BRISQUE, NIQE, and PIQE) were assessed alongside the proposed DE measure. The standard NR-IQAs were shown to fail at assessing image quality in the context of scanning, as the scanned images show a better score than their unscanned counterparts (see Table 1). DE, on the other hand, consistently resulted in better scores for the unscanned ground truth images compared to the scanned versions, as predicted. This shows DE to be a valid and reliable measure of scanned image quality.

### 3.2. Comparative Results

To assess the quality of output images from each scanning app (CleanPage, Microsoft Office Lens and Google PhotoScan), a variety of scan scenarios were tested. To allow comparisons to reference images, color and text images were synthesized and scanned. To test real world conditions, text on paper was scanned. Finally, to demonstrate the robustness to surface imperfections, scans were taken of content on crumpled paper. In each case, approximately fifty scans were taken. Visual results presented in Figure 3, Figure 4, Figure 5 and Figure 6 are sample images from each test set. For scoring using DE and SSIM, testing was done across each full dataset with the mean score is presented in each case. Sample results from each scan type, using each scanning app, are presented here.

CleanPage creates a homogenous background, crops to background surface automatically, and preserves image content while erasing surface imperfections. This creates a cleaner, more accurate reproduction of the original and therefore a better scan, while the other methods make for better photographs.

### 3.3. Image Quality Assessment

All scans were scored across the full dataset of approximately fifty subjects in each case using DE and SSIM. The original unscanned versions of the synthesized color and text image datasets were also included in the tests for reference. While visual results may be subjective, DE and SSIM offer objective ways of assessing the scans. Table 2 shows the DE scores for the three methods with the original, unscanned versions showing the best score as expected. The effects of real paper on the quality of the output can be seen clearly in the DE scores for the paper tests. 

SSIM is a measure of similarity but is normally used for measuring effects of noise, processing, distortion, etc. on an existing image, and so a ground truth of the same size, aspect ratio, and scale is assumed. This is not the case for scanned images, as each scanning method can suffer from alignment and resizing issues. SSIM has also been shown to fail at identifying color-based distortions [42], such as those introduced by scanning. As a result, SSIM is not a reliable indicator of the percentage similarity in this case. However, as a comparative measure, it is still useful for ranking the images in order of similarity to a reference, as it will still give the highest score to the most similar image. It is also helpful as a complement to the DE, which tests the consistency of the background surface, while the SSIM tests the retention of image content. The high SSIM scores coupled with low DE scores show that CleanPage retains image content while improving image quality. Table 3 shows the SSIM scores when each method’s scans are compared to the original unscanned images. 

### 3.4. User Experience

In the field of User Experience (UX), the golden rule is “the simpler the better”. Previous work in the area provides insight into the document scanning problem from a UX perspective, focusing on hand-grip diversity and showing how different users approach the problem [43]. Using this insight, the simplest possible UI was designed: one button. 

On tapping CleanPage’s one button, all phases are performed automatically, and an output image is produced. Since a sequence of images is captured for processing, any imperfections in user performance (jitter, defocusing, inconsistent distance, etc.) have less impact on the result, so capture does not require a particularly steady hand. CleanPage does not need a distinction to be made between image types, as it works equivalently for all capture.

In the other methods tested—Microsoft Office Lens, Google PhotoScan, and others—the result is highly dependent on the quality of the one image captured.

Microsoft Office Lens has the next simplest UI, requiring just a steady capture to produce the output image, after first selecting the input type (document, whiteboard, etc.).

Google’s scanner proved the trickiest to use. Once the capture button is pressed, four stitching points appear, and the camera must be moved to each point in order to capture the subject. This often leads to tilt warnings, as it is required that the camera be held at the same angle throughout. The user performance at this stage has a significant impact on the quality of results and both stitching errors (see Figure 4c) and cropping errors (see Figure 6b) are frequent.

#### Processing Time

Although scanning time depends highly on phone quality, a comparative analysis was made by using the same Android smartphone with all three scanning apps. Time was measured from the first step of the process in each case (see Figure 7, Figure 8 and Figure 9) to the appearance of the output image onscreen.

With CleanPage, a short sequence of images is captured (taking less than 1 s), and the simple methods used in each phase of the design mean processing is fast. The total capture time, although dependent on phone quality, is short. As there is no need to wait for a steady capture, the process is the same for every capture, so the processing time also remains more consistent.

Without manual intervention, Microsoft Office Lens’ scanning times were comparable, although capture time increases in poor lighting, where the bounding box can struggle to attach to the subject. Even without manual intervention, Google PhotoScan was the slowest, as four images need to be carefully captured. It also frequently required manual correction or repeated attempts, as cropping or stitching errors were common. This further increased the overall capture time, but these manual interventions were not included in the results in Table 4 for fair comparison.

## 4. Discussion

### 4.1. Scanning Performance Assessment

While scanning is becoming a day-to-day necessity as we move from physical workspaces to online environments, there is still much room for improvement in the technologies available and in the means of assessing these technologies. For example, the testing of accepted NR-IQAs highlighted their limitations when applied to scanned images, revealing that they are not suitable for this application. 

Along with a new high-quality scanning technology, two novel performance assessments are presented here: the discrete entropy as a quality measure and a new technique for testing using synthesized reference images. The discrete entropy is validated as it gives consistently better scores for unscanned ground truths than for their scanned counterparts. The new method of testing scanning performance, by using synthesized images displayed on a tablet, allows comparisons to be made to these ground-truth images using the standard reference-based quality measure, SSIM.

### 4.2. CleanPage

The quality of CleanPage’s outputs can be seen in both the visual results and in the qualitative and quantitative comparisons presented here with the scanning apps provided by software giants Microsoft and Google. Its fast and simple UX, without the need for manual intervention, makes it easier to use than other apps, particularly those using well-established stitching and mosaicking approaches. The design also reduces reliance on the user’s careful, steady hand to produce quality outputs, as it is relatively robust to motion. As seen in the presented results, it works on a variety of scan types without the need for user input, making it consistently fast across numerous scanning trials. The performance of the system is not only comparable to the leading scanning apps but outperforms them in removing surface imperfections to produce a clean scan, which was scored using DE, and in content retention, which was scored using SSIM, with synthesized ground-truth images as reference. 

In future work, a more thorough quality measure will be designed to capture contrast, smoothness, sharpness, etc. This quality measure may be used as an input to the system outlined here to select the best images in the sequence, remove poor quality images, and create a weighted average in the merging phase. While the focus here is on acquisition, the outputs from CleanPage could be combined with an OCR algorithm to develop a full scan-to-text design. Although initial testing shows consistency in OCR results, achieving typically less than 5% word error rate, OCR was not included here, as CleanPage is more broadly applicable to both document and whiteboard capture as well as to both text and images.

## Figures and Tables

**Figure 1 jimaging-06-00102-f001:**
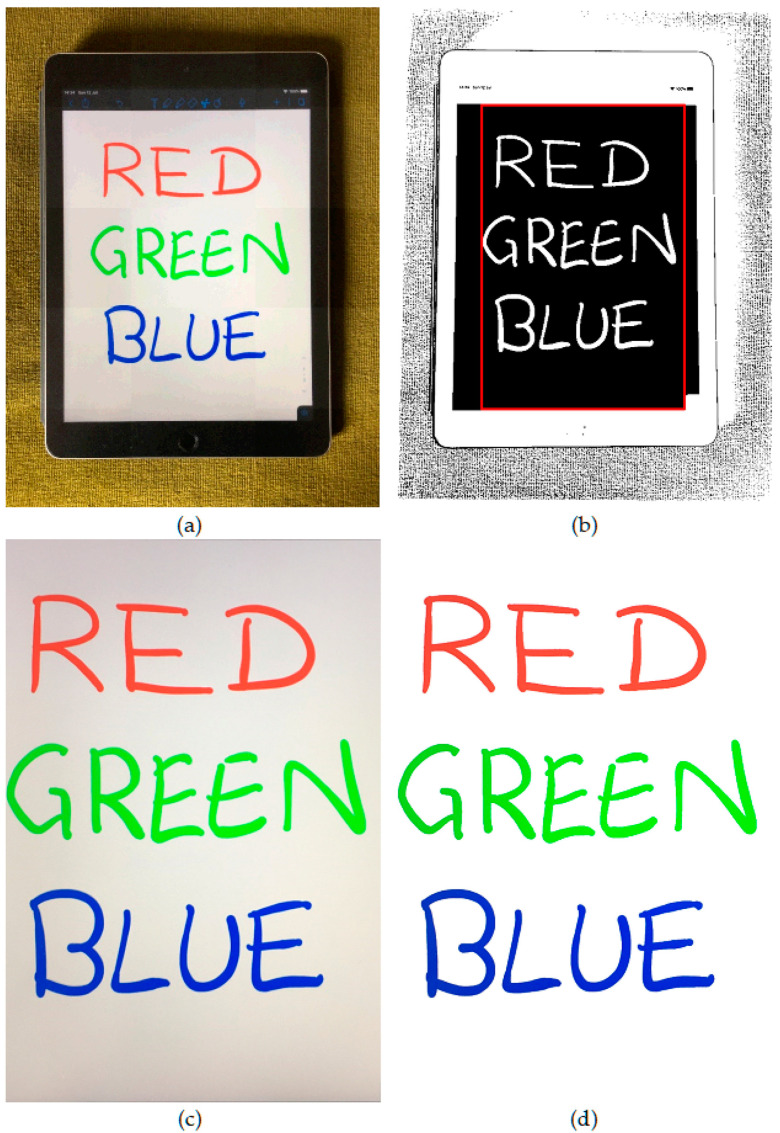
Stages of CleanPage. (**a**) A sample image from the original image sequence; (**b**) the binary surface mask showing the border search box; (**c**) the merged image after border removal; (**d**) after enhancement.

**Figure 2 jimaging-06-00102-f002:**
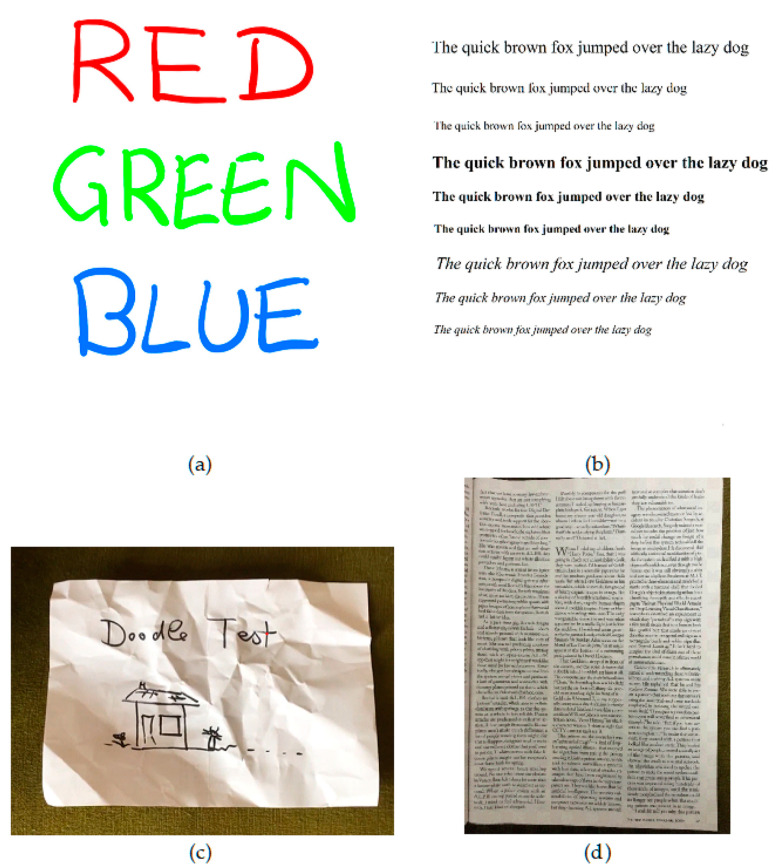
Sample test images from each dataset. (Note: real paper images are shown as photographs here). (**a**) A synthesized color reference image; (**b**) a synthesized text reference image; (**c**) a sketch on crumpled paper; (**d**) a sample of text on a page.

**Figure 3 jimaging-06-00102-f003:**
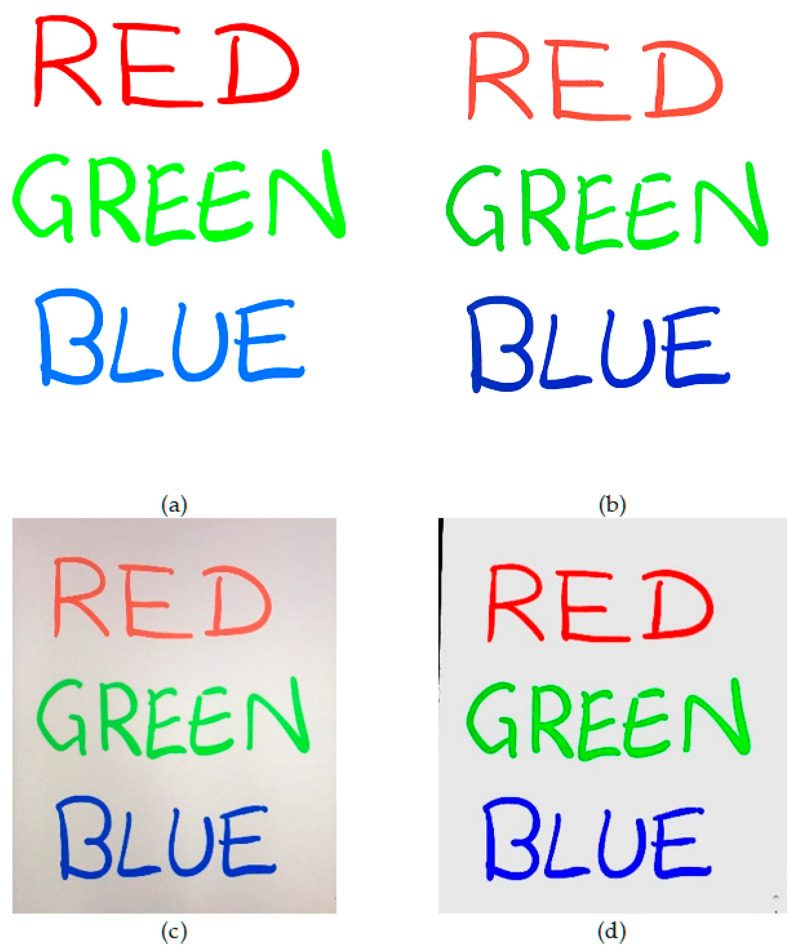
Sample results of an image from the synthesized color image dataset, scanned using a smartphone with the image showing on a tablet screen. (**a**) Original Image; (**b**) CleanPage; (**c**) Google PhotoScan; (**d**) Microsoft Office Lens.

**Figure 4 jimaging-06-00102-f004:**
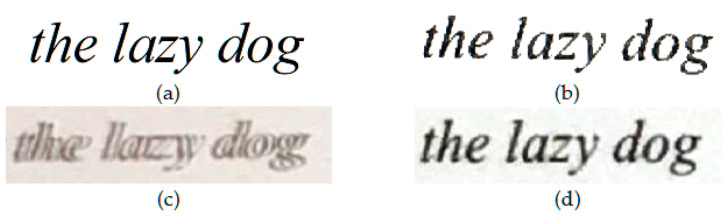
Zoomed results extracted from the smallest line of an image from the synthesized text image dataset, scanned using a smartphone with the image showing on a tablet screen. (**a**) Original Image; (**b**) CleanPage; (**c**) Google PhotoScan; (**d**) Microsoft Office Lens.

**Figure 5 jimaging-06-00102-f005:**
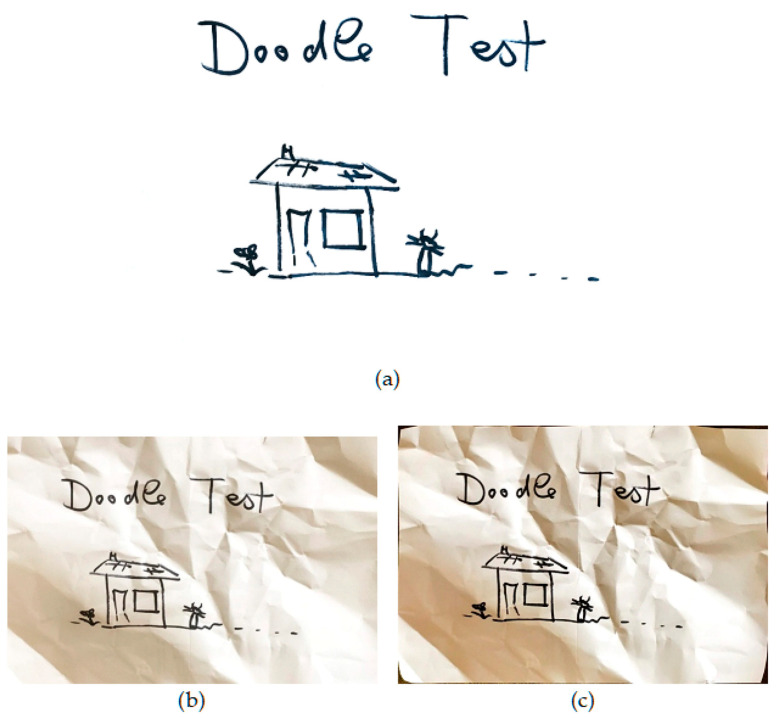
Sample results from scans of content on crumpled paper. (**a**) CleanPage; (**b**) Google PhotoScan; (**c**) Microsoft Office Lens.

**Figure 6 jimaging-06-00102-f006:**
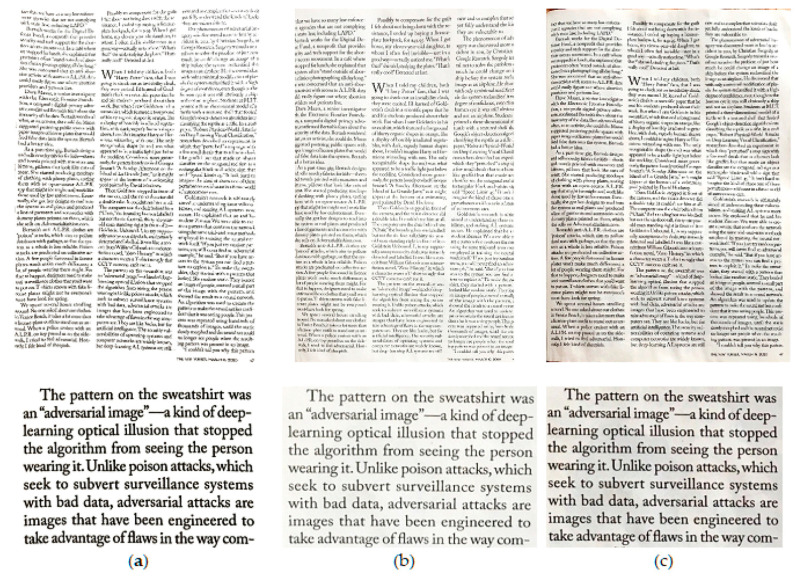
Sample results from a magazine page scan showing a zoomed paragraph. (**a**) CleanPage; (**b**) Google PhotoScan; (**c**) Microsoft Office Lens.

**Figure 7 jimaging-06-00102-f007:**
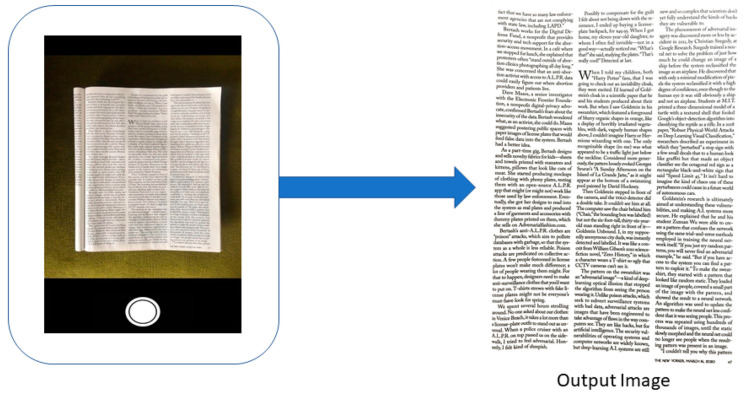
CleanPage’s simple user interface (UI). One button-tap results in the output image.

**Figure 8 jimaging-06-00102-f008:**
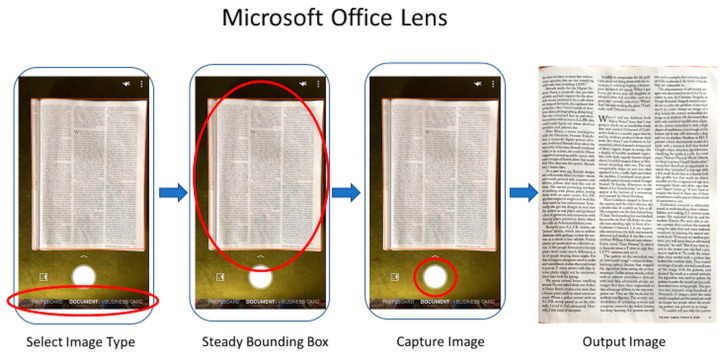
Microsoft Office Lens UI. An optional manual corner correction phase is also offered but not shown, as it is often not necessary and still yields good cropping results.

**Figure 9 jimaging-06-00102-f009:**
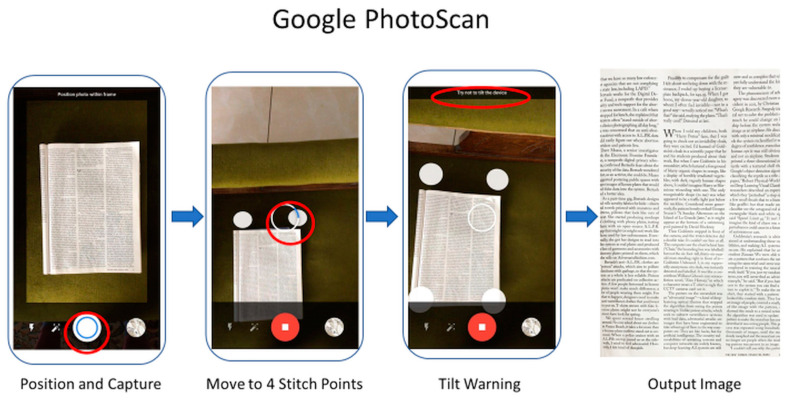
Google PhotoScan UI. An optional manual corner correction phase is also offered and is not shown, although is often necessary as automatic cropping is frequently incorrect.

**Table 1 jimaging-06-00102-t001:** Comparison of image quality measures using original unscanned reference images as ground truths (note: higher values denote lower quality). Scores are measured as the average across the full dataset of approximately fifty images in each case. Highlighted in red is where the measure erroneously gives a worse value to the better image (the original unscanned version).

Scoring Metric	Synthesized Color Image Dataset		Synthesized Text Image Dataset	
	Scanned	Original	Scanned	Original
BRISQUE	45.1108	**44.773**	**41.7858**	46.322
NIQE	**4.2445**	11.8021	**3.6031**	14.68
PIQE	**68.7442**	90.9064	**45.8493**	81.4116
DE	1.5917	**0.5978**	5.8915	**0.2743**

**Table 2 jimaging-06-00102-t002:** Comparison of scanning methods using discrete entropy (DE) as a quality measure. In each case, approximately fifty subjects were scanned using each method to produce the four datasets listed. The results presented are the average score across each dataset. A lower DE score suggests better quality.

Image Set	Original	CleanPage	Microsoft Office Lens	Google PhotoScan
**Synthesized Color Dataset**	**0.5978**	0.8941	1.5917	6.2870
**Synthesized Text Dataset**	**0.2743**	0.5319	5.8915	6.7118
**Crumpled Paper Dataset**	-	**0.5604**	7.2129	6.7611
**Text on Paper Dataset**	-	**1.7964**	6.7725	6.2016

**Table 3 jimaging-06-00102-t003:** Comparison of scanning methods using Structural Similarity score (SSIM) to compare each scan to synthesized reference images. Results presented are the average score across each full dataset. Higher scores indicate greater similarity to the original unscanned image.

Image Set	CleanPage	Microsoft Office Lens	Google PhotoScan
**Synthesized Color Dataset**	**0.8332**	0.7873	0.6170
**Synthesized Text Dataset**	**0.8602**	0.6963	0.7490

**Table 4 jimaging-06-00102-t004:** Comparison of scanning times over several trials. Time was measured from the first step of the process to the appearance of the output image onscreen.

Speed	CleanPage	Microsoft Office Lens	Google PhotoScan
**Scanning Time**	**8–11 s**	8–15 s	18–23 s

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
