# Peer review of "CleanPage: Fast and Clean Document and Whiteboard Capture"

_2313-433X, 2020, doi:10.3390/jimaging6100102_

Round 1

Reviewer 1 Report

1-Please, test your image with document corrupted with 5% or 10% noise rate

2-Please, what is your contribution in this paper, particularly in image enhancement and denoising

Author Response

Comment 1: 1-Please, test your image with document corrupted with 5% or 10% noise rate

 Response: As this app is scanning real subjects, it is not possible to do the standard quality versus noise level tests, as noise cannot be introduced during the capture process. This is also true of the scanning apps used for comparison (Google and Microsoft’s offerings). However, the discrete entropy has been shown previously to increase with noise distortion in text-based images [40] so CleanPage’s noise reduction can be observed in its low DE measures, as well as in the visual results.

Comment 2: 2-Please, what is your contribution in this paper, particularly in image enhancement and denoising

 Response: The novel contributions in “CleanPage: Fast and Clean Document and Whiteboard Capture” are as follows:

  • The CleanPage design: combining images from a short sequence using image features and homography for alignment, a background surface mask for cropping to content and the Yen threshold for contrast enhancement.
  • An assessment of accepted NR-IQAs, showing their limitations when applied to scanned images.
  • The use and validation of the discrete entropy as a quality measure for scanned images.
  • A comparative analysis of the leading scanning systems across a variety of different scan types.
  • A new testing method, using synthesized reference images as ground-truths in scanning assessment.

Reviewer 2 Report

The paper presents a method for capturing paper documents and whiteboard images using a smartphone. The method is quite straightforward and the sequence of operations is based on existing techniques. The only novelty that I remarked in this work is the use of multiple images (instead of one image and/or video). The work is an interesting and useful application. The paper is well written and the experimental results are convincing.

However, I didn’t see how and where the proposed method corrects the perspective distortion? In camera capture, this kind of distortion happens frequently.

Also, I understand that it is not possible to benchmark this method on existing datasets of camera capture documents. But I encourage the authors to make their captures publicly available, so that the research community may make use of them.

Author Response

The paper presents a method for capturing paper documents and whiteboard images using a smartphone. The method is quite straightforward and the sequence of operations is based on existing techniques.

Comment 1: The only novelty that I remarked in this work is the use of multiple images (instead of one image and/or video).

Response: Besides the use of multiple images, there are several novel contributions in this paper:

  • The CleanPage design: combining images from a short sequence using image features and homography for alignment, a background surface mask for cropping to content and the Yen threshold for contrast enhancement.
  • An assessment of accepted NR-IQAs, showing their limitations when applied to scanned images.
  • The use and validation of the discrete entropy as a quality measure for scanned images.
  • A comparative analysis of the leading scanning systems across a variety of different scan types.
  • A new testing method, using synthesized reference images as ground-truths in scanning assessment.

The work is an interesting and useful application. The paper is well written and the experimental results are convincing.

Comment 2: However, I didn’t see how and where the proposed method corrects the perspective distortion? In camera capture, this kind of distortion happens frequently.

Response: This is a great idea as perspective distortion is indeed an issue, particularly in handheld camera capture. However, this design is intended to compete with existing scanning apps where perspective correction is not included. It would definitely be a great addition in future work. A mention has been added to section 2.1.2, lines 151-153, to acknowledge this potential. 

Comment 3:   Also, I understand that it is not possible to benchmark this method on existing datasets of camera capture documents. But I encourage the authors to make their captures publicly available, so that the research community may make use of them.

Response: This is no problem at all, and the full datasets can be made available on publication.

Reviewer 3 Report

The paper presents the developed CleanPage mobile application and the methods used for image acquisition and processing. Although, most of the methods used by the Author are well-known, their application in the proposed software is novel and provide promising results.

The presented method and developed software have a potential for further improvements, being considered as an advantage, since it leads to good results at the current stage of its development.

The use of the full-reference image quality assessment methods such as SSIM cannot be expected as leading to ggod results, mainly due to the necessary image scaling but the Author is aware of their limitations and the presented motivation is convincing, also for NR-IQA methods.

Since the obtained image may be considered as textureless in the perfect case, an interesting element could the the comparison of the proposed application of entropy with some other texture analysis methods (e.g. some other Haralick features based on the GLCM - e.g. correlation or homogeneity).

The main drawback of the paper is the number of test images. To provide a reliable comparison with some other methods, a larger dataset of images should be used (captured by the same smartphone in the same controlled lighting conditions using three different applications). For example is is unclear if the results presented in Table 2 have been obtained for a single image or as the average values for more samples.

Since the proposed approach utilizing some small translations of consecutive video frames is very similar to the super-resolution methods (see e.g. Irani-Peleg approach - CVGIP, May 1991), they should also be mentioned and briefly presented in introduction.

Minor remarks:

  • there are some articles "a" left at the ends of text lines, e.g. line 162 or 127
  • the word "Gaussian" should be capitalized (line 135)
  • the word "Android" should be capitalized (line 335)
  • the word "averaged" (line 159) is a bit confusing and inappropriate considering the use of medians (as stated in line 157)

Although the presented approach is interesting and the results are very promising, in my opinion more detailed experimental results should be provided before the acceptance of the paper. SOme more details related to capturing of images should be provided as well (in fact the same image or set of images should be used in all applications but the problem is with different methods of their acquisition used by each of three applications).

Nevertheless, the technical quality of the paper can be considered as high.

Author Response

The paper presents the developed CleanPage mobile application and the methods used for image acquisition and processing. Although, most of the methods used by the Author are well-known, their application in the proposed software is novel and provide promising results.

The presented method and developed software have a potential for further improvements, being considered as an advantage, since it leads to good results at the current stage of its development.

Comment 1: The use of the full-reference image quality assessment methods such as SSIM cannot be expected as leading to ggod results, mainly due to the necessary image scaling but the Author is aware of their limitations and the presented motivation is convincing, also for NR-IQA methods.

Response: It is frankly refreshing to have a reviewer fully read and understand a paper to this extent. You have made my day! This is exactly so. The reason for including SSIM is that the discrete entropy alone won’t notice if content is lost. I’m working on developing a more complete scanned-image quality measure but in the meantime, the use of SSIM ensures that the DE results are trustworthy. A sentence has been added at lines 313-316 to capture this.

Comment 2: Since the obtained image may be considered as textureless in the perfect case, an interesting element could the the comparison of the proposed application of entropy with some other texture analysis methods (e.g. some other Haralick features based on the GLCM - e.g. correlation or homogeneity).

Response: Another excellent spot. I did look into texture analysis methods, but they tended to get confused between content and background in text-based images. I still think there is something in this, so watch this space for future work on the DE and texture analysis…

Comment 3: The main drawback of the paper is the number of test images. To provide a reliable comparison with some other methods, a larger dataset of images should be used (captured by the same smartphone in the same controlled lighting conditions using three different applications). For example is is unclear if the results presented in Table 2 have been obtained for a single image or as the average values for more samples.

Response: This is just poor presentation on my part. Each of the four image types tested consisted of datasets of approximately fifty subjects. Numerical results are the mean scores across these extensive tests. Results were generally consistent, so the mean sufficed for representation. The visual results shown are just sample images from each dataset. This has been clarified in section 2.2 and throughout the tables and results. It was felt that presenting more visual results would take away from the readability of the paper and would make it excessively long, but some more sample images from each dataset could be included if required.

Comment 4: Since the proposed approach utilizing some small translations of consecutive video frames is very similar to the super-resolution methods (see e.g. Irani-Peleg approach - CVGIP, May 1991), they should also be mentioned and briefly presented in introduction.

Response: The original version of CleanPage’s design actually used video and it is indeed very similar to super-resolution, so this has been added to the introduction in section 2 line 82 and reference [29].

Minor remarks:

  • there are some articles "a" left at the ends of text lines, e.g. line 162 or 127

Response: These are run-on sentences but have been checked throughout.

  • the word "Gaussian" should be capitalized (line 135)

Response: Poor old Gauss – and he’s one of my favourites! He’s all sorted now (line 136).

  • the word "Android" should be capitalized (line 335)

Response: Ditto. Android has been given the respect it deserves (line 349).

  • the word "averaged" (line 159) is a bit confusing and inappropriate considering the use of medians (as stated in line 157)

Response: This has been removed (line 162).

Although the presented approach is interesting and the results are very promising, in my opinion more detailed experimental results should be provided before the acceptance of the paper. Some more details related to capturing of images should be provided as well (in fact the same image or set of images should be used in all applications but the problem is with different methods of their acquisition used by each of three applications).

Response: See response to Comment 3 above. This detailed experimentation has been performed already, I just failed to get this across in the paper. I hope it is clearer now. Throughout testing, the same subjects have been used for all applications but, as you say, these cannot be the same exact images due to their different capture methods. However, the novel testing method proposed (using synthesized images) allows a certain workaround for this issue.

Nevertheless, the technical quality of the paper can be considered as high.

Response: As is the quality of this review.

Round 2

Reviewer 1 Report

Nothing

Reviewer 3 Report

The paper has been improved and the doubts mentioned in the previous review have been dispelled. Since the most relevant issue is related to the number of images used in experiments, the explanations added in the paper make the paper clear enough.

In my opinion the answers and are fully convincing and the corrections are satisfactory. The only minor issue is the copyrignt sign introduced automatically by text editor instead of "(c)" in Figures 3 and 4.